# Autochthonous Natural Starter Cultures: A Chance to Preserve Biodiversity and Quality of Pecorino Romano PDO Cheese

**Luigi Chessa** \*, **Antonio Paba, Elisabetta Daga, Ilaria Dupré, Carlo Piga, Riccardo Di Salvo, Martino Mura, Margherita Addis** and **Roberta Comunian**

Agris Sardegna, Agenzia Regionale per la Ricerca in Agricoltura, Associated Member of the JRU MIRRI-IT, Loc. Bonassai, SS291 km 18.600, 07100 Sassari, Italy; apaba@agrisricerca.it (A.P.); edaga@agrisricerca.it (E.D.); idupre@agrisricerca.it (I.D.); cpiga@agrisricerca.it (C.P.); rdisalvo@agrisricerca.it (R.D.S.); muramartino87@gmail.com (M.M.); maddis@agrisricerca.it (M.A.); rcomunian@agrisricerca.it (R.C.)
\* Correspondence: lchessa@agrisricerca.it

**Abstract:** During Pecorino Romano PDO cheese production, *scotta* (residual whey from *ricotta* cheese manufacturing) or *siero* (whey) can be integrated with autochthonous starters, natural and composed of an indefinite number of species and strains, or commercial selected starters to obtain *scotta/siero-innesto*. In this study, three biodiverse autochthonous natural starter cultures (SR30, SR56, and SR63) belonging to the Agris Sardegna BNSS microbial collection, composed of different strains belonging to the species *Streptococcus thermophilus*, *Lactobacillus delbrueckii* subsp. *bulgaricus*, *Enterococcus faecium*, and *Limosilactobacillus reuteri* were lyophilized and combined into two starter mixes (A and B). The *scotta/siero-innesto* and the Pecorino Romano obtained using natural starters were compared with those obtained using commercial selected starters during three seasons of the cheesemaking campaign. Different pH and microbial compositions for the *scotta/siero-innesto* obtained using natural or commercial starters were found, attributable to their different biodiversity. The six-month-ripened cheese microbiota was influenced mostly by the season of cheesemaking, whereas physico-chemical and sensory analyses did not highlight differences among the products obtained. In general, no effect attributable to the type of *scotta/siero-innesto* used was observed, allowing the conclusion that natural starter cultures can be used also in industrial-scale production, ensuring high stability in the technological performances and preserving the microbial, chemical, and sensory characteristics of Pecorino Romano PDO cheese.

**Keywords:** natural culture; selected starter; technological performances standardisation; cheese microbiology; physico-chemical characteristics; sensory analyses; Pecorino Romano PDO cheese; microbial collection; microbial biodiversity

## 1. Introduction

Natural microbial starters, constituted by lactic acid bacteria, commonly used to speed up the fermentation processes, have recently been supplanted by the commercial selected ones in order to ensure high stability in the technological performances and compensate for the microbial load reduction due to the improvement of the hygienic milking and cheese-making conditions [1,2]. However, the use of commercial starters is responsible for a decrease in the microbial biodiversity of the dairy environment and final products, such as traditional and protected designation of origin (PDO) cheeses [3]. In fact, commercial starters are constituted by a small number of selected species and strains, whereas natural starters consist of undefined microbial consortia, with different strains for each species, hanging in a flexible balance [4]. Moreover, due to their biodiversity, natural cultures are better adapted to the raw material to be processed [5], supposed to be able to confer sensory richness to the product [3], and less sensitive to phage attacks responsible for failures in fermentation [6]. Nevertheless, the main drawback attributable to natural starters is the inconstancy of technological performances, due to their variability in microbial

load and composition over time. For this reason, in recent years, the Pecorino Romano PDO producers expressed the need to have, at their disposal, effective starter cultures such as *scotta-innesto* (a natural deproteinised whey starter culture) but in an easy-to-use form (lyophilised), while respecting the characteristics of autochthony and biodiversity requested by the product specification [7,8]. Among the mandatory accomplishments, the preparation of the microbial starter is one of the key activities. The residual whey obtained from the *ricotta* cheese manufacturing (*scotta*) must be incubated at 42–45 °C for 18–20 h in order to obtain the *scotta-innesto* [9] to be used as indirect inoculum in sheep's milk for the next Pecorino Romano cheesemaking. Prior to incubation, *scotta* can be integrated with starter cultures in order to have adequate microbial development [8]. *Siero* (whey), instead of *scotta*, can be used for the preparation of the microbial inoculum for Pecorino Romano. At the moment, the use of only a limited number of lyophilised commercial starter cultures, made up of a few autochthonous but selected strains included in a restricted access list (http://pecorino.ineq.it, accessed on 24 June 2021), is allowed as integration for the obtainment of *scotta-innesto*. However, to have complex natural starters available in freeze-dried form, to supplement the *scotta*, would be preferable. In this study, three autochthonous natural starter cultures, preserved at the Agris Sardegna laboratories since the 1960s, were reproduced in a lyophilised and ready-to-use form. A multidisciplinary microbiological, physico-chemical, and sensory approach was applied to investigate their technological performances and stability, in comparison with commercial starters, during experimental manufactures of Pecorino Romano PDO cheese.

## 2. Materials and Methods

### 2.1. Experimental Plan

The technological performances of three natural starter cultures (SR30, SR56, and SR63) belonging to the Agris Sardegna BNSS microbial collection (www.mbds.it, accessed on 24 June 2021), collected in the 1960s from local dairy plants located in Berchidda (Sardinia Italy) and previously characterised for their microbial composition [10] and biodiversity at the strain-level [11], were investigated in this study. The three cultures, combined in order to obtain the starter mixes MixA and MixB (indicated as A and B, respectively), both having the same cocci/bacilli ratio (3:1) [12], were used as indirect inoculum for cheese manufacturing trials, at the industrial scale, in comparison with the starters usually used in each of the dairy plants, as control (C).

Cheesemaking was performed at three industrial dairy plants in Sardinia (indicated as Factory 1, 2, and 3) during three different seasons of the cheesemaking year (January, March, and June). The microbial development of both natural and commercial selected starter cultures, used for the preparation of the *scotta/siero-innesto* (SiA, SiB, and SiC) was investigated by classical and molecular techniques. Since at Factory 3, only two fermenters were available, no SiC was prepared, and lyophilised C culture was added directly to milk. In each of the three dairy plants, in three seasons of the dairy year, three replicates of the cheesemaking were performed for each season, for a total of nine experimental cheesemaking days, using the three types of starters (SiA, SiB, and SiC) on each cheesemaking day. During cheese manufacturing, the microbial growth and acidification performances of natural and commercial selected starters were compared, as well as the microbiological, physico-chemical, and sensory characteristics of the cheeses after six months of ripening. Analyses for the monitoring of the microbial evolution were performed during all the cheesemaking steps, whereas physico-chemical and sensory analyses were performed only on six-month-ripened cheeses (CA, CB, and CC).

### 2.2. Large Scale Production of Freeze-Dried Natural Cultures

The biodiverse cultures SR30, SR56, and SR63, previously collected in toto, characterised for their microbial composition at the strain level [11] and stored in lyophilised form, were reactivated in sterile *scotta* (Alimwhey, Alimenta S.r.l., Cagliari, Italy) at 42 °C for 18 h, concentrated by centrifugation at $8000\times g$ for 20 min and frozen. Then, the cultures

were shipped to Bioagro S.r.l. (Thiene, Italy) for large-scale production in freeze-dried form. The lyophilised cultures were mixed and packaged in order to obtain the cultures A (SR30 + SR56) and B (SR30 + SR63) described by Chessa et al. [12].

### 2.3. Cheesemaking and Sampling

*Scotta* or *siero* (the latter only in Factory 3) were used as substrates for the growth of natural (A and B) or control starter cultures (C). The commercial starters used in each dairy plant to integrate *scotta* or *siero*, for control starter cultures production, were approved by the Control Plan PDO cheese, as indicated by the Inspection Body [13]. On each cheesemaking day, uninoculated *scotta/siero* (US), obtained from previous Pecorino Romano cheese manufacturing, was transferred into three fermenters and heated at 80 °C for 10 min. After cooling down to 42° C, two fermenters were inoculated with the freeze-dried natural starters A or B, respectively, in order to obtain a microbial inoculum of 3.7 Log CFU/mL. The third fermenter was inoculated with the commercial starter (C) at the dose (g) usually applied by each cheese factory, based on the experience of the cheesemaker, and then the microbial inoculum (Log CFU/mL) was determined by microbial plate counts (see below). After mechanical agitation, the *scotta/siero* was incubated at 42 °C for 18 h for the preparation of the *scotta/siero-innesto* (SiA, SiB, and SiC).

Before each experimental cheese manufacturing, milk was thermised (Supplementary Table S1), transferred into three multi-purpose circular vats of 6000 to 8000 L size, inoculated, respectively, with SiA, SiB, and SiC, and then coagulated with commercial calf rennet paste (35–40 g/100 L of milk) (Caglificio Manca, Thiesi, Italy). Technological parameters such as clotting time and curd firmness time, temperature, time and size of curd cutting, and other technological parameters adopted in each dairy plant were reported in the Supplementary Material (Table S1). After cutting, the cheese wheels were transferred into special moulds and incubated at 30–45 °C for 20–25 min, and the pH was monitored for 24 h at room temperature. Heating temperature and time, as well as the pH after heating, varied depending on the protocols applied in each dairy plant (Table S1). The experimental cheeses (CA, CB, and CC) were ripened for 6 months in the respective dairy plants at 12–14 °C and 80–85% relative humidity.

Ripened cheeses were aseptically sampled at the dairy plants and immediately transferred, under refrigeration, to the laboratory for microbial, chemical, and sensory analyses. Moreover, microbial analysis was also performed on *scotta/siero* before and immediately after the inoculum, *scotta/siero-innesto*, and uninoculated and inoculated thermised milk. For molecular analysis, aliquots of 1 mL each (two replicates) of *scotta/siero-innesto* were stored at −80 °C until analysis.

### 2.4. Microbial Counts

The concentration of total bacteria, thermophilic cocci and bacilli, enterococci, and mesophilic lactobacilli was assessed by plate counts in uninoculated *scotta/siero* (US); *scotta/siero* inoculated with A, B, and C cultures (SA, SB, and SC); *scotta/siero-innesto* (SiA, SiB, and SiC); thermised milk (M); milk inoculated with the *scotta/siero-innesto* (MA, MB, and MC); and cheeses (CA, CB, and CC) at 6 months of ripening. Total bacterial counts were performed on MPCA (Microbiol, Cagliari, Italy) and incubated at 42 °C for 48 h in aerobiosis. Thermophilic cocci were enumerated on M17 agar (Microbiol) at 37 °C for 48 h aerobically; thermophilic bacilli on MRS agar pH 5.4 (Microbiol) at 42 °C for 48 h anaerobically using Oxoid™ AnaeroGen™ (Thermo Fisher Scientific, Waltham, MA, USA); enterococci on KAA agar (Microbiol) at 42 °C for 48 h aerobically; mesophilic lactobacilli on FH agar [14] at 37 °C for 72 h anaerobically; citrate-fermenting bacteria on modified MRS agar [15] at 37 °C for 72 h anaerobically; and coliforms on VRBA MUG at 37 °C for 18 h aerobically.

In thermised milk and six-month-ripened cheese, the most probable number (MPN) of presumptive propionic acid bacteria and clostridia, in the Buti medium incubated at

37 °C for 7 days, was also evaluated. For clostridia enumeration, samples were pre-treated at 80 °C for 15 min.

Microbial counts were expressed as average values ± standard deviation (SD) Log CFU/mL or Log CFU/g.

### 2.5. Molecular Quantification and Detection of Starter Lactic Acid Bacteria (SLAB) Species

Total community DNA was extracted from samples of the *scotta/siero-innesto* obtained from A, B, and C cultures used in the three cheese factories, following the protocol described by Paba et al. [16]. The determination of the concentration of *Streptococcus thermophilus*, *Lactobacillus delbrueckii* subsp. *lactis*, and *L. delbrueckii* subsp. *bulgaricus* by real-time quantitative PCR (qPCR) was based on the quantification of the relative abundance (target gene per 16S rRNA gene (*rrn*) copies) of the genes *lacZ*, *prtH*, and *dpp*E, respectively, using the primers described by Cremonesi et al. [17] and the protocol described by Paba et al. [16]. The presence of *L. helveticus* was investigated by PCR in 24 μL MegaMix™ (Gel Company, Inc., San Francisco, CA, USA), using 0.5 μL of each primer [17] and an FTA Disc for DNA analysis (GE Healthcare, Chicago, IL, USA) as a template.

### 2.6. Chemical Analysis

Experimental cheeses, at 6 months of ripening, were analysed for pH (pH-meter Crison Basic 20+); dry matter (DM) [18]; moisture, calculated as follows: moisture = (100- DM), fat [19], and total nitrogen (TN) [20]; protein calculated as follows: protein = [(TN × 6.38]; and sodium chloride by potentiometric titration with AgNO$_3$ (88:2006]) (automatic titrator Mettler-Toledo DL55, Mettler-Toledo GmbH, Schwerzenbach, Switzerland). Proteolysis was evaluated in terms of Proteolytic Indices: soluble nitrogen (SN) at pH 4.6, soluble nitrogen in 12% trichloroacetic acid (TCA-SN), and soluble nitrogen in 10% phosphotungstic acid (PTA-SN) [21]. Lipolysis was determined by measuring individual and total free fatty acids (FFAs) content as previously described by Addis et al. [22].

### 2.7. Sensory Analysis

The Difference From Control test [23] method for the quantification of sensory differences between cheeses A or B and the cheese C, used as control, was applied. Twenty-four trained panellists rated the size of the difference between each sample (A and B) and the control (C) on a 9-point scale (1 = no difference, 9 = very large difference). Subjects were informed that some of the test samples could have been the same as the control. In fact, a blind control was included in the test samples, in order to estimate the "placebo" effect, induced by asking the difference question when no difference exists. All the samples were labelled with a 3-digit number code and served to subjects in a completely balanced randomized design order. Subjects were also instructed to clean their mouth with unsalted crackers and water before testing each sample. The difference between sample and control was rated separately for each of the three sensory characteristics: odour, taste, and texture.

### 2.8. Statistical Analysis

Kinetics of curd acidification were processed by Prism (v. 7; GraphPad Software, La Jolla, CA, USA), and linear regression was used for the curves' slopes calculation and comparison by the Student's *t*-test ($P < 0.05$).

Microbial counts for the evaluation of the inoculum in *scotta/siero* inoculated with A, B, and C, and milk inoculated with the *scotta/siero-innesto* SiA, SiB, and SiC, in different cheesemaking seasons were statistically investigated by the one-way analysis of variance (ANOVA). Differences between the individual means (SA, SB, or SC, and MA, MB, and MC, respectively) were compared by the Tukey–Kramer post hoc test ($P < 0.05$) using SPSS Statistics (v. 21.0; IBM Corp., Armonk, NY, USA). For microbial counts of *scotta/siero-innesto* and six-month-ripened cheeses, and for molecular quantification of lactic acid bacteria by qPCR in *scotta/siero-innesto*, the effects of the starter culture used, the season of

cheesemaking, and the interaction culture $\times$ season were evaluated by the general linear model (GLM), using SPSS Statistics.

Chemical parameters such as gross composition, proteolytic indices, and FFAs in cheese were statistically investigated to verify the effects of the starter culture (F, 3 levels, A, B, and C), the season of cheesemaking (F, 3 levels, January, March, and June), and the interaction culture $\times$ season using a general linear model (GLM), by Minitab 16 (Minitab 16 Statistical Software, 2010, State College, PA, USA, Minitab, Inc.). The comparison among means was performed using ANOVA Tukey–Kramer post hoc test ($P < 0.05$).

Different from control test data were statistically evaluated by ANOVA and the Dunnett's multiple comparisons post hoc test, using R Software (v.3.4.1, https://www.r-project.org, accessed on 24 June 2021).

## 3. Results

### 3.1. Technological Parameters

The experimental cheesemaking was performed according to the Pecorino Romano PDO cheese technology, in three different dairy plants (Factories 1, 2, and 3) and three seasons of the production year (January, March, and June). Thermised milk was inoculated with the *scotta/siero-innesto* SA, SB, and SC, and coagulated at 38–39 °C using lamb rennet paste. The clotting time and the curd firmness time ranged between 9–12 min and 15–18 min, respectively. After cutting, the curd was cooked at 46–48 °C, then mechanically pressed and stewed at 30–42 °C for 12–25 min.

The pH values measured for the *scotta/siero-innesto* obtained from the natural starter cultures A and B did not differ from each other ($P < 0.05$) and ranged from 3.9 to 4.1, whereas the control *scotta-innesto* obtained with the commercial selected starter C was, in general, more acid (pH, 3.4 and 3.8) ($P < 0.05$), especially in Factory 1 (Table 1).

**Table 1.** pH measured in *scotta/siero-innesto*.

| Factory | SiA | SiB | SiC |
|---------|-----|-----|-----|
| 1 | 3.9 ± 0.1 [a] | 4.0 ± 0.1 [a] | 3.4 ± 0.1 [b] |
| 2 | 4.0 ± 0.1 [a] | 4.0 ± 0.0 [a] | 3.8 ± 0.1 [b] |
| 3 | 4.1 ± 0.0 [a] | 4.1 ± 0.0 [a] | n.d. |

For each factory, average pH values (± standard deviation) of the *scotta/siero-innesto* SiA, SiB, and SiC, obtained after incubation of *scotta/siero* inoculated with A, B, or C, respectively, sharing the same superscript letters (where present) did not differ significantly ($P < 0.05$), according to the Tukey–Kramer HSD post hoc test. n.d., not determined.

The curds acidifications were comparable in the progression of pH monitored in the first 20 h and revealed no significant ($P < 0.05$) differences, depending on the *scotta/siero-innesto* used (Figure 1).

### 3.2. Microbial Counts

The enumeration of total bacteria, thermophilic cocci and lactobacilli, enterococci, and mesophilic lactobacilli in uninoculated and inoculated *scotta/siero*, in *scotta/siero-innesto*, in milk inoculated with *scotta/siero-innesto*, and in cheese are reported in the following subparagraphs.

#### 3.2.1. Uninoculated and Inoculated *Scotta/Siero*

Total bacterial counts in uninoculated *scotta/siero* (US) were about 2 Log CFU/mL, except in samples from Factory 3 in March (4.1 Log CFU/mL) (Figure 2). Moreover, generally, only thermophilic cocci and lactobacilli were detected in US (Figure 2), but at Factory 3, where *siero* was used instead of *scotta*, also enterococci and mesophilic lactobacilli were detected in March (Figure 2). The microbial inoculum in *scotta/siero* SA and SB in Factories 1 and 2 was always constant (about 3.75 Log CFU/mL) and significantly ($P < 0.05$) lower than in the *scotta/siero* inoculated with the control commercial starter SC (about 5 and 7 Log CFU/mL, respectively) following the Factories' practices (Figure 2). No SC was

prepared in Factory 3, and the commercial selected starter C was added directly to the milk (see paragraph 3.2.3).

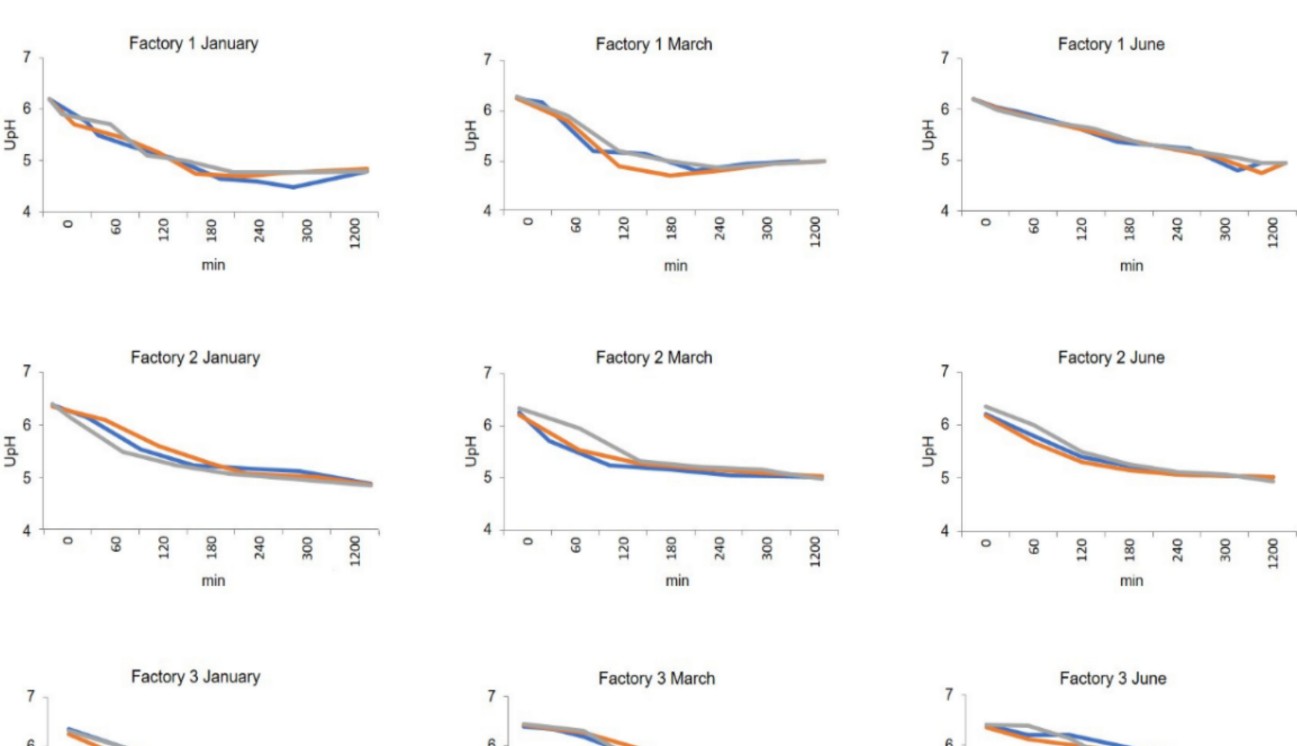

**Figure 1.** Acidification curves of Pecorino Romano PDO curds, measured in each of the three factories (1, 2, and 3) in the three seasons of the cheesemaking year (January, March, and June), obtained using the *scotta/siero-innesto* A, B, and C, during the first 1200 min after cheesemaking. Average acidification values were expressed as UpH.

### 3.2.2. *Scotta/Siero-Innesto*

After incubation at 42 °C for 18 h, microbial counts in *scotta/siero-innesto* were performed. No significant ($P < 0.05$) differences in the concentration of total bacteria, thermophilic cocci, and lactobacilli among the cultures (A, B, and C) in each season were observed, except for slight variations in the concentration of enterococci (Figure 3). Mesophilic lactobacilli were detected only in *siero-innesto* from Factory 3, during the March and June trials. The effects of the starter culture used, the season of cheesemaking, and the interaction culture × season were evaluated by GLM analysis (Table 2). The composition of the microbial consortia highlighted significant ($P < 0.001$) differences in the type of starter culture used in Factories 1 and 2, but no effects of season or of interaction culture × season were observed. The differences were mostly attributable to the concentration of enterococci, found in SiA and SiB but absent in SiC, and also to thermophilic cocci and bacilli, especially in Factory 1 (Figure 3 and Table 2). No significant effects were observed in the microbial composition of the *siero-innesto* in Factory 3, obtained using only the cultures A and B, since no SiC was prepared (Figure 3), and lyophilised C culture was added directly to milk.

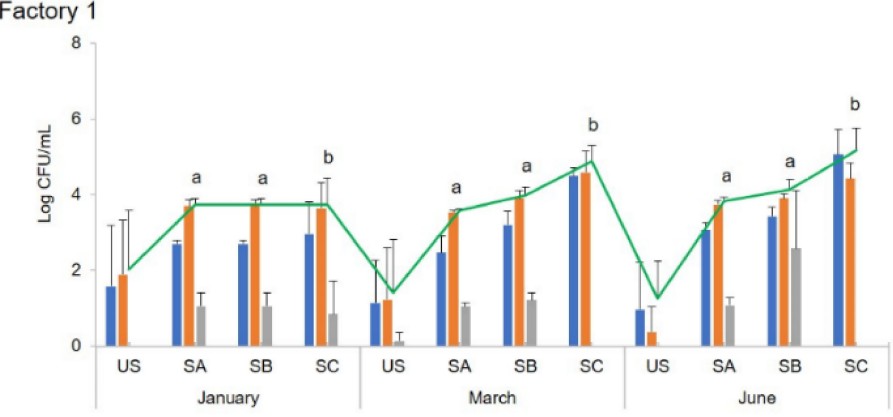

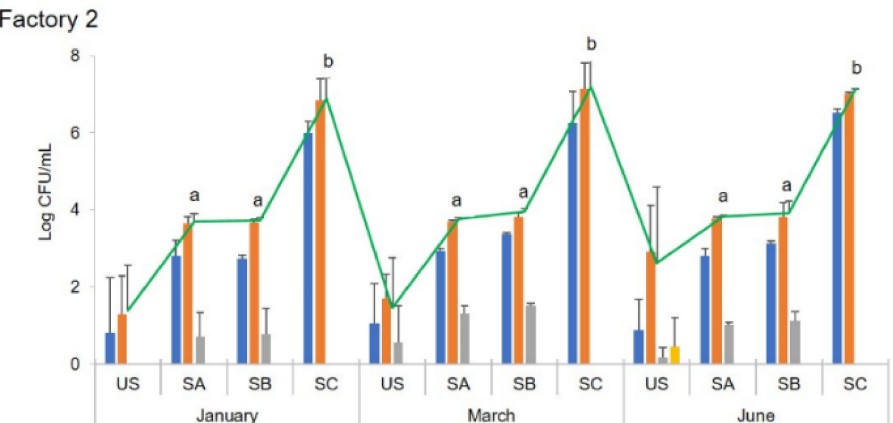

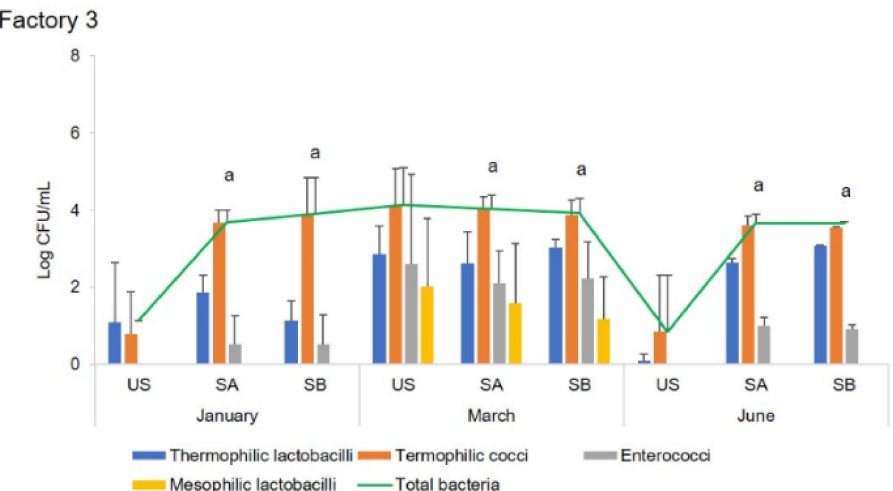

**Figure 2.** Microbial counts of thermophilic lactobacilli, thermophilic cocci, enterococci, mesophilic lactobacilli, and total bacteria in uninoculated *scotta/siero* (US), and in *scotta/siero* inoculated with A, B, and the commercial starters (SA, SB, and SC) in the three Factories (1, 2, and 3) involved in the study during the three seasons of the cheesemaking year (January, March, and June). Microbial counts were expressed as Log CFU/mL ± standard deviation. For each factory and cheesemaking period, total bacterial counts in SA, SB, and SC sharing the same apex letters did not differ significantly ($P < 0.05$), according to the Tukey–Kramer HSD post hoc test.

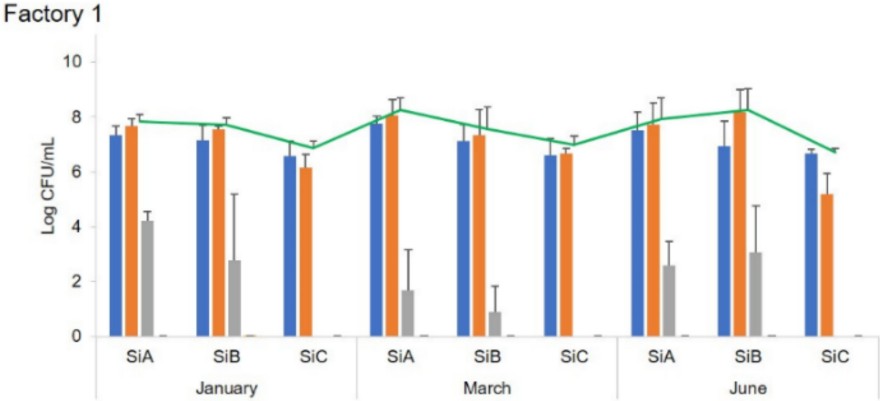

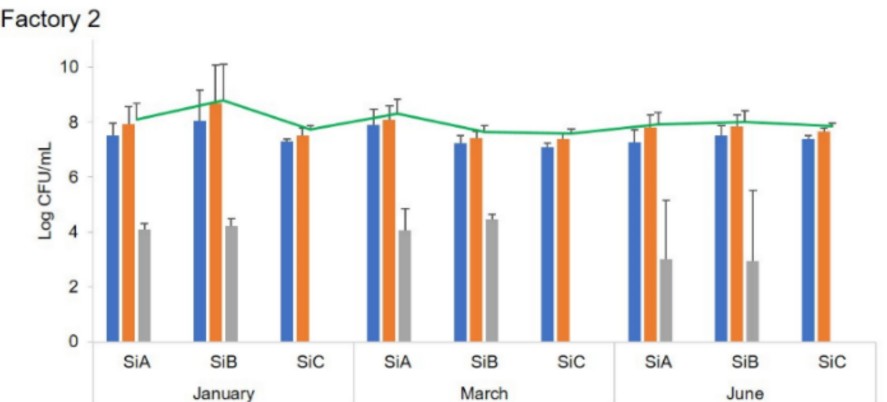

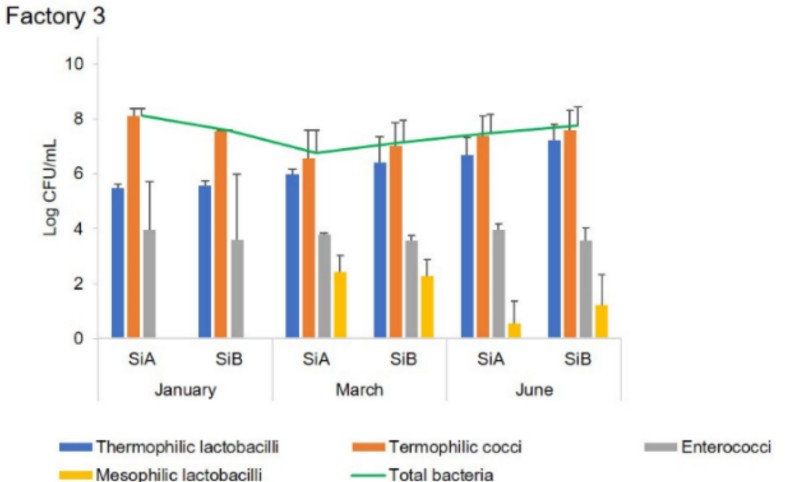

**Figure 3.** Microbial counts of thermophilic lactobacilli, thermophilic cocci, enterococci, mesophilic lactobacilli, and total bacteria in *scotta/siero-innesto* obtained using natural or commercial starters (SiA, SiB, or SiC), in the three factories (1, 2, and 3) involved in the study, during the three seasons of the cheesemaking year (January, March, and June). Microbial counts were expressed as Log CFU/mL $\pm$ standard deviation. The statistical evaluation of the effects of culture, season, and culture $\times$ season on the microbial development of *scotta/siero-innesto*, performed by the general linear model (GLM), was reported in Table 2.

**Table 2.** Statistical evaluation of culture and/or season effects on the development of starter cultures in *scotta/siero-innesto*.

| Factory | Microbial Group | Culture | Season | Culture × Season |
|---|---|---|---|---|
| 1 | Total microbial population | *** | n.s. | n.s. |
| | Thermophilic lactobacilli | * | n.s. | n.s. |
| | Thermophilic cocci | *** | n.s. | n.s. |
| | Mesophilic bacteria | n.s. | n.s. | n.s. |
| | Enterococci | *** | * | n.s. |
| 2 | Total microbial population | *** | n.s. | n.s. |
| | Thermophilic lactobacilli | n.s. | n.s. | n.s. |
| | Thermophilic cocci | n.s. | n.s. | n.s. |
| | Mesophilic bacteria | n.s. | n.s. | n.s. |
| | Enterococci | *** | n.s. | n.s. |
| 3 | Total microbial population | n.s. | n.s. | n.s. |
| | Thermophilic lactobacilli | * | n.s. | n.s. |
| | Thermophilic cocci | n.s. | n.s. | n.s. |
| | Mesophilic bacteria | n.s. | n.s. | n.s. |
| | Enterococci | n.s. | n.s. | n.s. |

The statistical effect of the microbial culture used (A, B, or C), the cheesemaking season (January, March, or June), and the interaction culture × season on the development of the main microbial groups and on total microbial population in *scotta/siero-innesto* were evaluated by general linear model (GLM) analysis in the three factories (1, 2, and 3) involved in the study. *, $P < 0.05$; ***, $P < 0.001$; n.s., not significant.

### 3.2.3. Thermised and Inoculated Milk

In thermised milk (M), total microbial concentration ranged between 2.34 and 3.57 Log CFU/mL, with the exceptions of Factory 2 in March and June (5.06 and 5.37 Log CFU/mL, respectively) and Factory 3 in June (4.09 Log CFU/mL) (Figure 4). M was inoculated with the *scotta/siero-innesto* obtained from each of the three cultures (MA, MB, and MC) and, in each dairy plant for each cheesemaking period, the level of the microbial inoculum in milk was similar; indeed, no significant ($P < 0.05$) differences were found among MA, MB, and MC (Figure 4). In Factory 3, the lyophilised starter C was added directly to milk, but no differences in the microbial inoculum were found (Figure 4).

### 3.2.4. Cheese

The enumeration of total bacteria, thermophilic cocci and lactobacilli, enterococci, and mesophilic lactobacilli in six-month-ripened cheeses was carried out (Figure 5). The effect of the starter culture used, the season of cheesemaking, and their interaction were evaluated by GLM analysis and shown in Table 3. Microbial counts revealed a similar concentration of thermophilic lactobacilli and cocci and mesophilic lactobacilli in CA, CB, and CC cheeses, whereas variable concentrations of enterococci and citrate-fermenting bacteria were observed (Figure 5). In fact, significant ($P < 0.001$) effects of starter culture and/or cheesemaking season on the concentration of enterococci were observed in Factories 1 and 2 by GLM analysis (Table 3). Moreover, in Factory 1, significant ($P < 0.05$) interaction culture × season on the number of thermophilic lactobacilli, the main SLAB group enumerated in cheeses, was observed, while enterococci in Factory 2 were significantly influenced by the starter culture used. Citrate-fermenting bacteria were affected by season, in cheeses manufactured in Factories 2 and 3, whereas mesophilic lactobacilli were never affected by culture and/or season. However, total microbial consortia of cheeses from all three dairy plants were influenced only by the cheesemaking season but not by the starter culture used for the preparation of the *scotta/siero-innesto* nor by the interaction culture × season.

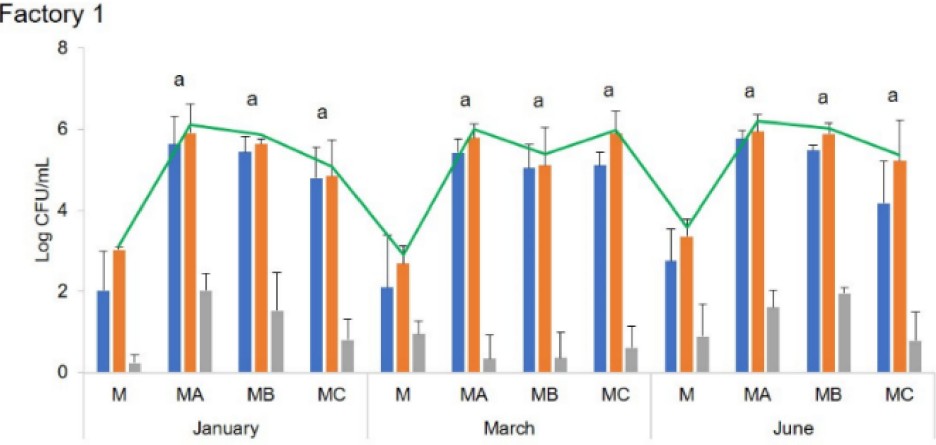

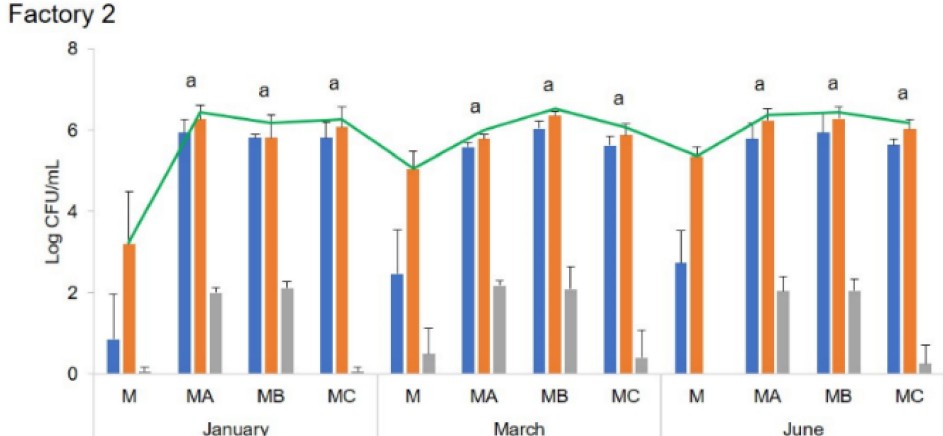

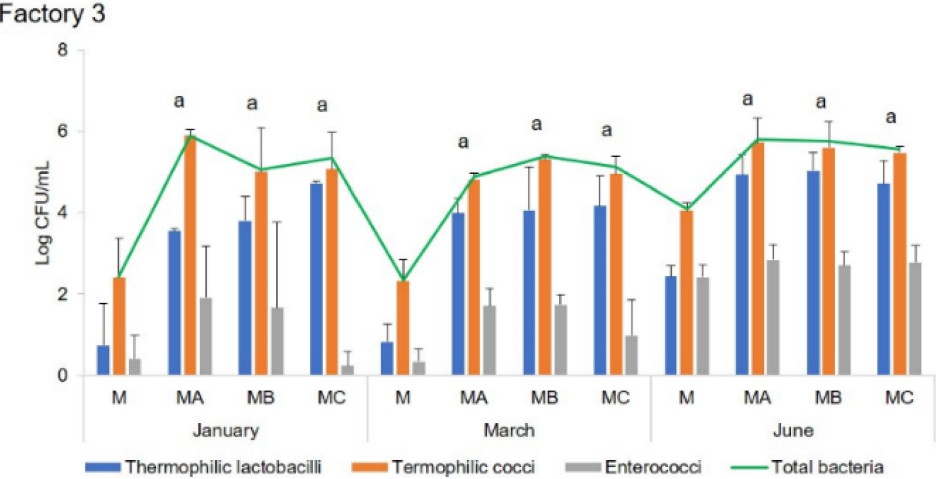

**Figure 4.** Microbial counts of thermophilic lactobacilli, thermophilic cocci, enterococci, and total bacteria in uninoculated milk (M) and in milk inoculated with the *scotta/siero-innesto* obtained from the three cultures A, B, and C (MA, MB, and MC) in the three factories (1, 2, and 3) involved in the study, during the three seasons of the cheesemaking year (January, March, and June). Microbial counts were expressed as Log CFU/mL ± standard deviation. For each factory and cheesemaking season, total bacterial counts in MA, MB, and MC sharing the same apex letters do not differ significantly ($P < 0.05$), according to the Tukey–Kramer HSD post hoc test.

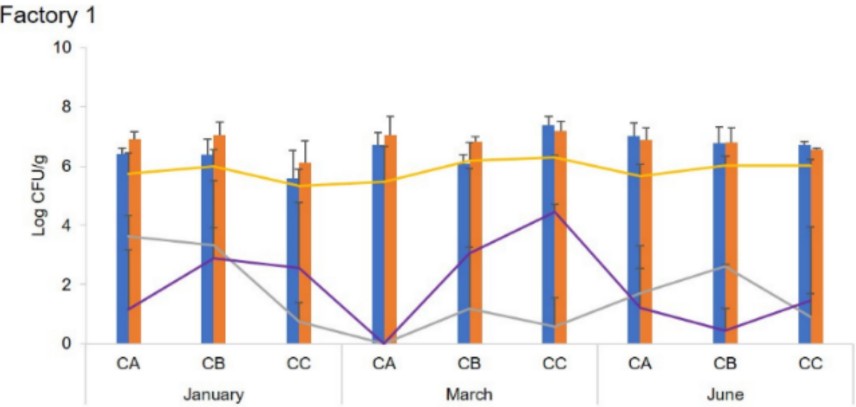

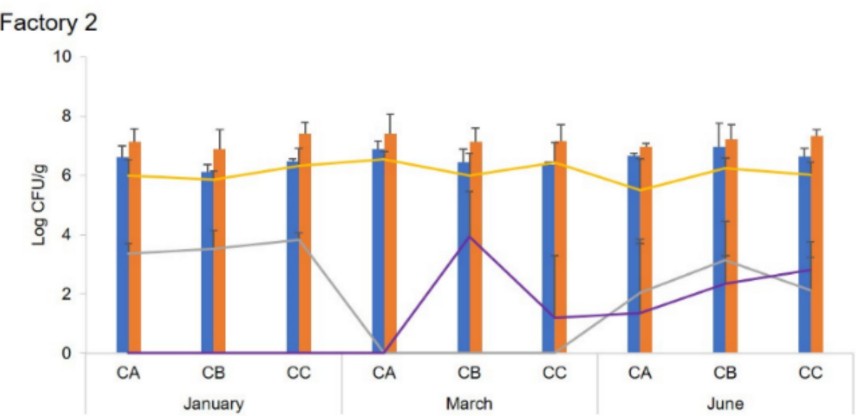

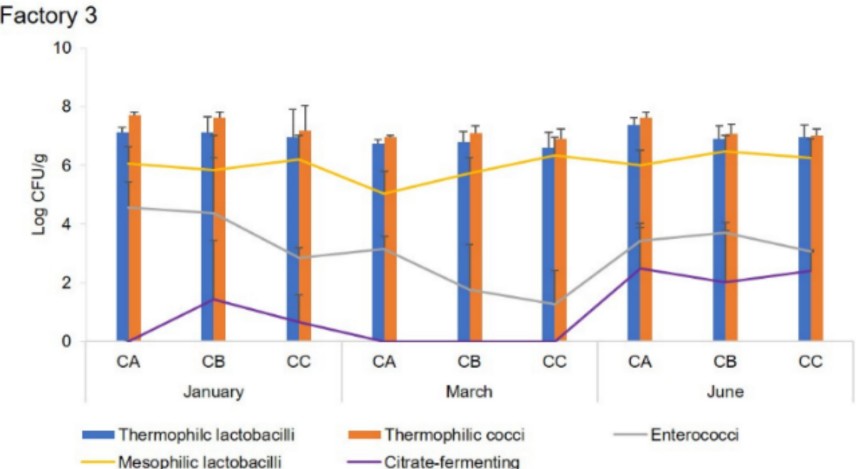

**Figure 5.** Microbial counts of thermophilic lactobacilli, thermophilic cocci, enterococci, mesophilic lactobacilli, and citrate-fermenting bacteria in six-month-ripened cheeses obtained using the three *scotta/siero-innesto* (CA, CB, or CC), in the three Factories (1, 2, and 3) involved in the study, during the three periods of the cheesemaking year (January, March, and June). Microbial counts were expressed as Log CFU/g ± standard deviation. The statistical evaluation of the effects of culture, season, and culture × season on the cheese microbiota, performed by the general linear model (GLM), was reported in Table 3.

### 3.3. Molecular Quantification and Detection of SLAB Species

The concentration of *S. thermophilus*, *L. delbrueckii* subsp. *lactis*, and *L. delbrueckii* subsp. *bulgaricus* in *scotta/siero-innesto* was investigated through the quantification of the relative abundances of species-specific genes by real-time qPCR (Figure 6). No effect of the culture

(A, B, or C) used for the preparation of *scotta/siero-innesto* was observed for any of the microbial species tested, whereas a significant ($P < 0.05$) effect of the cheesemaking season on the abundance of all the above-mentioned microbial species was found by GLM analysis (Table 4). The interaction culture × season was not significant in all the factories (Table 4). *L. helveticus* was not detected in any of the SiA and SiB collected in all the three factories, while it was found in all the SiC from Factory 1, and only in SiC collected in January from Factory 2 (data not shown).

**Table 3.** Statistical evaluation of culture and/or cheesemaking season effects on the main microbial groups in cheeses at 6 months of ripening.

| Factory | Microbial Group | Culture | Season | Culture × Season |
|---|---|---|---|---|
| 1 | Total microbial population | n.s. | ** | n.s. |
| | Thermophilic lactobacilli | n.s. | * | * |
| | Thermophilic cocci | n.s. | n.s. | n.s. |
| | Mesophilic lactobacilli | n.s. | n.s. | n.s. |
| | Enterococci | *** | *** | n.s. |
| | Citrate fermenting | n.s. | n.s. | n.s. |
| 2 | Total microbial population | n.s. | *** | n.s. |
| | Thermophilic lactobacilli | n.s. | n.s. | n.s. |
| | Thermophilic cocci | n.s. | n.s. | n.s. |
| | Mesophilic lactobacilli | n.s. | n.s. | n.s. |
| | Enterococci | n.s. | *** | n.s. |
| | Citrate fermenting | n.s. | * | n.s. |
| 3 | Total microbial population | n.s. | *** | n.s. |
| | Thermophilic lactobacilli | n.s. | n.s. | n.s. |
| | Thermophilic cocci | n.s. | n.s. | n.s. |
| | Mesophilic lactobacilli | n.s. | n.s. | n.s. |
| | Enterococci | n.s. | n.s. | n.s. |
| | Citrate fermenting | n.s. | ** | n.s. |

The statistical effects of the microbial culture used (A, B, or C), the cheesemaking season (January, March, or June), and the interaction culture × season on the main microbial groups and total microbial population in six-month-ripened cheeses were evaluated by general linear model (GLM) analysis in the three factories (1, 2, and 3) involved in the study. *, $P < 0.05$; **, $P < 0.01$; ***, $P < 0.001$; n.s., not significant.

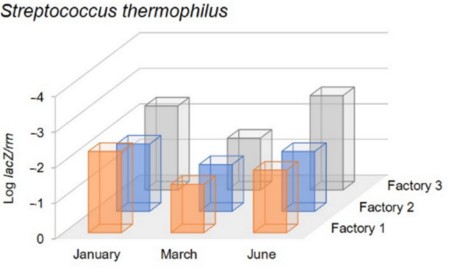

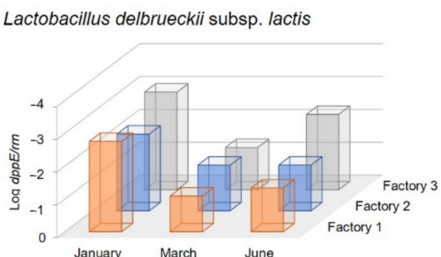

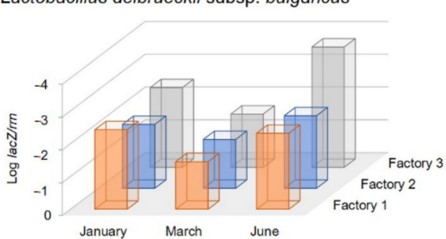

**Figure 6.** Relative abundance (species-specific gene/*rrn*) of *Streptococcus thermophilus*, *Lactobacillus delbrueckii* subsp. *lactis*, and *L. delbrueckii* subsp. *bulgaricus* in *scotta/siero-innesto* investigated by qPCR, in the three factories (1, 2, and 3) involved in the study during the three seasons of the cheesemaking year (January, March, and June). For each microbial species, the relative abundances were expressed as Log of the species-specific gene/*rrn*.

**Table 4.** Statistical evaluation of culture and/or season effects on the molecular quantification of the main microbial species in *scotta/siero-innesto*.

| Microbial Specie | Factory | Culture | Season | Culture × Season |
|:---:|:---:|:---:|:---:|:---:|
| *S. thermophilus* | 1 | n.s. | * | n.s. |
| | 2 | n.s. | *** | n.s. |
| | 3 | n.s. | * | n.s. |
| *L. delbrueckii* subsp. *bulgaricus* | 1 | n.s. | * | n.s. |
| | 2 | n.s. | ** | n.s. |
| | 3 | n.s. | *** | n.s. |
| *L. delbrueckii* subsp. *lactis* | 1 | n.s. | *** | n.s. |
| | 2 | n.s. | *** | n.s. |
| | 3 | n.s. | *** | n.s. |

The statistical effect of the microbial culture used (A, B, or C), the cheesemaking season (January, March, or June), and the interaction culture × season on the abundance of *S. thermophilus*, *L. delbrueckii* subsp. *bulgaricus*, and *L. delbrueckii* subsp. *lactis* in *scotta/siero-innesto* were evaluated by general linear model (GLM) analysis in the three factories (1, 2, and 3) involved in the study. *, $P < 0.05$; **, $P < 0.01$; ***, $P < 0.001$; n.s., not significant.

### 3.4. Physico-Chemical Characterisation of Cheese

Physico-chemical parameters and gross composition of six-month-ripened experimental Pecorino Romano PDO cheeses, produced in the three factories (1, 2, and 3), were reported in Table 5. The effects of the starter cultures used, natural (A and B) or commercial selected (C) ones, and the season of cheesemaking (January, March, and June) were statistically evaluated. In general, no effect ($P < 0.05$) attributable to the type of *scotta/siero-innesto* used was observed on the main macro-composition parameters, except for moisture and NaCl content (Table 5). In Factories 1 and 2, moisture content was lower in cheeses produced with the culture C than in those produced with natural cultures, particularly in March and June. In Factory 1, cheeses produced in June with natural culture B showed a NaCl content significantly higher than other cheeses. Also in Factory 3, in June, a similar trend was found for CB cheeses, although NaCl concentration was not significantly different from CA and CC cheeses in the same cheesemaking season. The season of cheesemaking generally affected ($P < 0.05$) the gross composition parameters of cheeses. In particular, in Factory 2, cheeses produced in January always had a higher moisture content than those produced in March and June, regardless of the starter culture used. Cheeses produced in March (Factory 3) showed significantly lower fat content than those produced in January and June (Table 5).

The statistical evaluation of the overall effects of the culture (C), season (S), and culture x season were evaluated by a GLM (general linear model). The proteolytic indexes SN/TN, TCA-SN/TN, and PTA-SN/TN [21] revealed that cheeses obtained using natural cultures (CA and CB) generally showed a proteolytic process of ripening very similar to each other and comparable to that of the cheeses manufactured using commercial selected cultures (C). A lower proteolysis (although not significant) was measured in the control cheese (CC) compared to that produced using the natural cultures in March and June at Factory 2 (Table 6). CA cheeses produced in June at Factory 3 showed an NS/NT content significantly higher than CB and CC cheeses manufactured in the same season. The period of cheesemaking, in general, did not affect the proteolytic process of the examined cheeses.

Free fatty acids (FFAs) profiles of experimental Pecorino Romano cheeses were grouped into three classes: short-chain FFAs (SCFFAs), medium-chain FFAs (MCFFAs), and long-chain FFAs (LCFFAs) (Table 7). The most representative FFA for each class (C4:0, C16:0, and C18:1, respectively) and total FFAs (TFFAs) content were also reported. The different *scotta/siero-innesto* used, obtained from the three cultures (A, B, and C), did not affect the lipolytic ripening process of cheeses, which was, instead, weakly influenced by the cheesemaking season. Lipolysis was more pronounced in cheeses produced in January at Factory 1 (although not significantly) and at Factory 3.

**Table 5.** Physico-chemical and gross-composition parameters of experimental Pecorino Romano DOP cheeses (CA, CB, and CC).

**Factory 1**

| Season | January | | | March | | | June | | | Effect | | |
|---|---|---|---|---|---|---|---|---|---|---|---|---|
| Culture | CA | CB | CC | CA | CB | CC | CA | CB | CC | S | C | S×C |
| pH | 5.3 ± 0.1 ab | 5.4 ± 0.2 a | 5.3 ± 0.2 ab | 5.2 ± 0.02 ab | 5.2 ± 0.0 ab | 5.0 ± 0.0 ab | 4.9 ± 0.1 b | 5.0 ± 0.2 ab | 5.0 ± 0.3 ab | *** | ns | ns |
| Moisture (%) | 30.6 ± 0.4 a | 31.0 ± 1.0 a | 31.1 ± 0.3 a | 31.2 ± 0.2 a | 31.0 ± 1.0 a | 31.1 ± 0.2 a | 31.0 ± 1.0 a | 30.4 ± 0.1 a | 28.0 ± 1.0 b | *** | * | ** |
| Fat (%) | 35.4 ± 0.5 ab | 35.0 ± 1.0 ab | 35.0 ± 1 ab | 34.0 ± 0.3 ab | 33.8 ± 0.1 b | 33.9 ± 0.4 ab | 34.0 ± 1.0 ab | 35.0 ± 0.4 ab | 35.8 ± 0.4 a | *** | ns | ns |
| Protein (%) | 26 ± 1.0 | 26.0 ± 1.0 | 25.9 ± 0.2 | 26.2 ± 0.3 | 26.0 ± 1 | 26.0 ± 0.2 | 25.1 ± 0.1 | 25.0 ± 1.0 | 25.2 ± 0.4 | ns | ns | ns |
| NaCl (%) | 3.6 ± 0.3 b | 3.4 ± 0.3 b | 3.3 ± 0.4 b | 3.4 ± 0.2 b | 3.5 ± 0.2 b | 3.5 ± 0.1 b | 4.2 ± 0.1 b | 6.0 ± 1.0 a | 4.1 ± 0.2 b | *** | *** | *** |

**Factory 2**

| Season | January | | | March | | | June | | | Effect | | |
|---|---|---|---|---|---|---|---|---|---|---|---|---|
| Culture | CA | CB | CC | CA | CB | CC | CA | CB | CC | S | C | S×C |
| pH | 4.9 ± 0.1 ab | 4.9 ± 0.1 ab | 4.9 ± 0.2 ab | 5.1 ± 0.1 a | 5.1 ± 0.1 a | 5.0 ± 0.1 a | 4.9 ± 0.1 ab | 5.0 ± 0.2 ab | 4.6 ± 0.0 b | ** | ns | ns |
| Moisture (%) | 33 ± 1.0 a | 32.3 ± 0.5 ab | 32.0 ± 0.4 ab | 30 ± 1.0 cd | 31 ± 1.0 c | 29.1 ± 0.4 d | 31.4 ± 0.2 bc | 30.2 ± 0.2 cd | 30.5 ± 0.1 cd | *** | *** | * |
| Fat (%) | 33 ± 1.0 c | 33.5 ± 0.4 abc | 34 ± 1.0 abc | 33.4 ± 0.5 abc | 33 ± 1.0 bc | 34 ± 1.0 abc | 33.9 ± 0.2 abc | 35.0 ± 0.2 a | 34.5 ± 0.2 ab | ** | ns | ns |
| Protein (%) | 24.0 ± 0.3 b | 24.9 ± 0.1 ab | 25.1 ± 0.5 ab | 25.7 ± 0.4 a | 25 ± 1.0 ab | 25.7 ± 0.5 a | 25 ± 1.0 ab | 25.2 ± 0.3 ab | 25 ± 1.0 ab | ** | ns | ns |
| NaCl (%) | 5.0 ± 0.2 | 4.8 ± 0.5 | 5 ± 1.0 | 5.2 ± 0.5 | 5.3 ± 0.5 | 5 ± 1.0 | 5.5 ± 0.3 | 5.2 ± 0.4 | 5.6 ± 0.3 | ns | ns | ns |

**Factory 3**

| Season | January | | | March | | | June | | | Effect | | |
|---|---|---|---|---|---|---|---|---|---|---|---|---|
| Culture | CA | CB | CC | CA | CB | CC | CA | CB | CC | S | C | S×C |
| pH | 4.9 ± 0.1 | 4.9 ± 0.2 | 4.9 ± 0.0 | 5.1 ± 0.0 | 4.9 ± 0.0 | 5.0 ± 0.1 | 5.1 ± 0.1 | 4.9 ± 0.2 | 4.8 ± 0.1 | ns | * | ns |
| Moisture (%) | 31.0 ± 0.1 | 32.0 ± 1.0 | 31.0 ± 1.0 | 33.0 ± 1.0 | 33.0 ± 1.0 | 31.7 ± 0.2 | 31.9 ± 0.5 | 33.0 ± 1.0 | 31.6 ± 0.3 | ns | ns | ns |
| Fat (%) | 34.3 ± 0.2 a | 34.0 ± 1.0 ab | 34.0 ± 1.0 a | 31.1 ± 0.4 c | 31.0 ± 1.0 c | 31.8 ± 0.2 bc | 33.7 ± 0.5 ab | 32.0 ± 1.0 abc | 33.4 ± 0.5 ab | *** | ns | ns |
| Protein (%) | 25.3 ± 0.2 ab | 24.6 ± 0.4 ab | 25.0 ± 1.0 ab | 25.0 ± 1.0 ab | 26.0 ± 1.0 ab | 26.6 ± 0.5 a | 24.3 ± 0.5 b | 24.0 ± 1.0 b | 25.0 ± 0.3 ab | *** | ns | ns |
| NaCl (%) | 5.0 ± 1.0 | 5.0 ± 1.0 | 5.0 ± 1.0 | 5.2 ± 0.2 | 5.0 ± 1.0 | 4.9 ± 0.4 | 4.8 ± 0.4 | 6.0 ± 1.0 | 5.2 ± 0.5 | ns | ns | ns |

Physico-chemical and gross-composition parameters of experimental Pecorino Romano DOP cheese manufactured with natural (A and B) and commercial (C) cultures in three different seasons of the cheesemaking year (January, March, and June), in the three factories (1, 2, and 3) involved in the study. For each parameter and cheesemaking season, average values within rows (± standard deviation) sharing the different superscript letters differ significantly, according to the ANOVA Tukey–Kramer HSD post hoc test ($P < 0.05$). The statistical evaluation of the overall effects of culture (C), season (S), and culture x season (S×C) were evaluated by GLM (general linear model) analysis. * $P < 0.05$; ** $P < 0.01$; *** $P < 0.001$; ns, not significant.

**Table 6.** Proteolytic indices of experimental Pecorino Romano DOP cheeses (CA, CB, and CC).

**Factory 1**

| Season | | January | | | | | March | | | | | June | | | | | Effect | | |
|---|---|---|---|---|---|---|---|---|---|---|---|---|---|---|---|---|---|---|---|
| Culture | CA | | CB | | CC | | CA | | CB | | CC | | CA | | CB | | CC | | S | C | S×C |
| NS/NT | 14 ± 1 | | 15 ± 1 | | 16.4 ± 0.3 | | 15 ± 1 | | 14 ± 1 | | 14 ± 1 | | 15 ± 2 | | 15 ± 1 | | 15 ± 1 | ns | ns | ns |
| NS-TCA/NT | 12 ± 1 | | 13 ± 2 | | 14.6 ± 0.4 | | 13 ± 1 | | 12 ± 2 | | 13 ± 1 | | 13 ± 2 | | 12 ± 1 | | 12 ± 2 | ns | ns | ns |
| NS-PTA/NT | 10 ± 1 ab | | 10 ± 1 ab | | 11 ± 1 a | | 10 ± 1 ab | | 9 ± 3 ab | | 11 ± 1 a | | 10 ± 2 ab | | 7 ± 1 b | | 10 ± 2 ab | ns | ** | ns |

**Factory 2**

| Season | | January | | | | | March | | | | | June | | | | | Effect | | |
|---|---|---|---|---|---|---|---|---|---|---|---|---|---|---|---|---|---|---|---|
| Culture | CA | | CB | | CC | | CA | | CB | | CC | | CA | | CB | | CC | | S | C | S×C |
| NS/NT | 12 ± 1 | | 12 ± 1 | | 12 ± 1 | | 13 ± 1 | | 13 ± 1 | | 11 ± 1 | | 14 ± 1 | | 14 ± 1 | | 12 ± 2 | ns | ns | ns |
| NS-TCA/NT | 10 ± 1 | | 10 ± 1 | | 10 ± 2 | | 11 ± 1 | | 11 ± 1 | | 8 ± 2 | | 10 ± 2 | | 11.8 ± 0.4 | | 9 ± 3 | ns | ns | ns |
| NS-PTA/NT | 8 ± 1 | | 7 ± 1 | | 8 ± 2 | | 9 ± 1 | | 8 ± 1 | | 6 ± 2 | | 8 ± 2 | | 7.8 ± 0.2 | | 7 ± 3 | ns | ns | ns |

**Factory 3**

| Season | | January | | | | | March | | | | | June | | | | | Effect | | |
|---|---|---|---|---|---|---|---|---|---|---|---|---|---|---|---|---|---|---|---|
| Culture | CA | | CB | | CC | | CA | | CB | | CC | | CA | | CB | | CC | | S | C | S×C |
| NS/NT | 11.6 ± 0.5 ab | | 11.7 ± 0.2 ab | | 11 ± 1 b | | 12.3 ± 0.5 ab | | 13 ± 1 ab | | 12 ± 1 ab | | 14 ± 1 a | | 11 ± 1 b | | 11 ± 1 b | * | ns | * |
| NS-TCA/NT | 8.7 ± 0.5 ab | | 9.7 ± 0.2 ab | | 9 ± 1 ab | | 10 ± 1 ab | | 12 ± 1 a | | 10 ± 1 ab | | 11 ± 1 ab | | 8 ± 2 b | | 9.2 ± 0.5 ab | ns | ns | * |
| NS-PTA/NT | 7 ± 1 | | 7 ± 1 | | 8 ± 1 | | 9 ± 1 | | 9 ± 2 | | 9 ± 1 | | 8 ± 2 | | 6 ± 1 | | 7 ± 1 | ns | ns | ns |

Proteolytic indices of experimental Pecorino Romano DOP cheeses manufactured with natural (CA and CB) and control (CC) culture starters in three different periods of the dairy season (January, March, and June) in the three factories (1, 2, and 3) involved in the study. For each parameter and cheesemaking season, average values within rows (± standard deviation) sharing the different superscript letters differ significantly, according to the ANOVA Tukey–Kramer HSD post hoc test ($P < 0.05$). The statistical evaluation of the overall effects of the culture (C), season (S), and culture × season (S×C) were evaluated by GLM (general linear model) analysis. *, $P < 0.05$; **, $P < 0.01$; *** $P < 0.001$; ns, not significant.

**Table 7.** Individual and total free fatty (mmol/kg of cheese) of experimental Pecorino Romano DOP cheese (CA, CB, and CC).

**Factory 1**

| Season | | January | | | | | March | | | | | June | | | | | Effect | | |
|---|---|---|---|---|---|---|---|---|---|---|---|---|---|---|---|---|---|---|---|
| Culture | CA | | CB | | CC | | CA | | CB | | CC | | CA | | CB | | CC | | S | C | S×C |
| C4:0 | 27 ± 18 | | 34 ± 25 | | 29 ± 26 | | 5.9 ± 0 | | 5.6 ± 0.1 | | 7.1 ± 0.3 | | 7 ± 1 | | 7 ± 1 | | 8.1 ± 0.2 | ns | ns | ns |
| Short Chain | 32 ± 17 | | 39 ± 24 | | 34 ± 26 | | 10.7 ± 0.4 | | 10.5 ± 0.1 | | 12.6 ± 0.4 | | 12 ± 2 | | 12 ± 1 | | 13.4 ± 0.2 | ns | ns | ns |
| C16:0 | 2.2 ± 0.1 ab | | 2 ± 0.2 b | | 2.3 ± 0.1 ab | | 2.2 ± 0.2 ab | | 2.2 ± 0.1 ab | | 2.3 ± 0.2 ab | | 2.5 ± 0.2 a | | 2.5 ± 0.2 a | | 2.6 ± 0.2 a | *** | ns | ns |
| Medium Chain | 4.8 ± 0.1 ab | | 4.3 ± 0.3 b | | 4.7 ± 0.1 ab | | 4.8 ± 0.4 ab | | 4.9 ± 0.2 ab | | 5.1 ± 0.4 ab | | 5.3 ± 0.4 a | | 5.3 ± 0.5 a | | 5.5 ± 0.4 a | *** | ns | ns |
| C18:1 9 c | 1.7 ± 0 | | 1.7 ± 0.1 | | 1.8 ± 0 | | 1.8 ± 0.2 | | 1.8 ± 0.1 | | 1.8 ± 0.2 | | 1.9 ± 0.1 | | 1.96 ± 0.05 | | 2 ± 0.2 | ns | ns | ns |
| Long Chain | 2.7 ± 0.1 | | 2.6 ± 0.2 | | 2.7 ± 0.1 | | 2.8 ± 0.3 | | 2.6 ± 0.2 | | 2.7 ± 0.3 | | 2.8 ± 0.1 | | 3 ± 0.1 | | 3.1 ± 0.3 | ns | ns | ns |
| TFFA | 40 ± 17 | | 46 ± 25 | | 41 ± 26 | | 18 ± 1 | | 18 ± 0.4 | | 20 ± 1 | | 20 ± 2 | | 20 ± 2 | | 22 ± 0.5 | ns | ns | ns |

**Table 7.** *Cont.*

**Factory 2**

| Season | January | | | March | | | June | | | Effect | | |
|---|---|---|---|---|---|---|---|---|---|---|---|---|
| Culture | CA | CB | CC | CA | CB | CC | CA | CB | CC | S | C | S×C |
| C4:0 | 11 ± 1 | 11 ± 1 | 11 ± 1 | 9 ± 2 | 12 ± 8 | 9 ± 3 | 6 ± 1 | 6 ± 1 | 7 ± 1 | ns | ns | ns |
| Short Chain | 18 ± 2 | 19 ± 1 | 19 ± 1 | 17 ± 4 | 22 ± 14 | 16 ± 5 | 11 ± 1 | 10 ± 1 | 12 ± 2 | ns | ns | ns |
| C16:0 | 3.1 ± 0.5 | 3.1 ± 0.3 | 3.3 ± 0.3 | 3 ± 0.4 | 3 ± 2 | 3 ± 1 | 2.8 ± 0.3 | 2.7 ± 0.2 | 3.1 ± 0.3 | ns | ns | ns |
| Medium Chain | 6.7 ± 0.9 | 6.8 ± 0.6 | 7 ± 1 | 7 ± 1 | 8 ± 4 | 6 ± 1 | 5.7 ± 0.5 | 5.4 ± 0.4 | 6 ± 1 | ns | ns | ns |
| C18:1 9c | 2.6 ± 0.1 | 2.6 ± 0 | 2.7 ± 0.1 | 2.4 ± 0.5 | 3 ± 1 | 3 ± 1 | 2.2 ± 0.2 | 2.2 ± 0.2 | 2.6 ± 0.3 | ns | ns | ns |
| Long Chain | 4 ± 0.2 | 3.9 ± 0.1 | 4.2 ± 0.2 | 4 ± 1 | 5 ± 2 | 4 ± 1 | 3.4 ± 0.2 | 3.4 ± 0.2 | 3.9 ± 0.3 | ns | ns | ns |
| TFFA | 29 ± 3 | 30 ± 1 | 30 ± 2 | 27 ± 5 | 34 ± 21 | 26 ± 7 | 20 ± 1 | 18 ± 1 | 22 ± 3 | ns | ns | ns |

**Factory 3**

| Season | January | | | March | | | June | | | Effect | | |
|---|---|---|---|---|---|---|---|---|---|---|---|---|
| Culture | CA | CB | CC | CA | CB | CC | CA | CB | CC | S | C | S×C |
| C4:0 | 6 ± 2 [a] | 7 ± 1 [a] | 6 ± 1 [a] | 4 ± 0.5 [b] | 3.5 ± 0.2 [b] | 3.9 ± 0.4 [b] | 3.8 ± 0.1 [b] | 3.8 ± 0.2 [b] | 3.8 ± 0.1 [b] | *** | ns | ns |
| Short Chain | 14 ± 4 [a] | 14 ± 4 [a] | 14 ± 3 [a] | 9 ± 2 [ab] | 7 ± 1 [ab] | 9 ± 2 [ab] | 6.8 ± 0.4 [b] | 6.9 ± 0.2 [b] | 6.9 ± 0.2 [b] | *** | ns | ns |
| C16:0 | 3 ± 1 | 3 ± 1 | 3 ± 1 | 2.1 ± 0.4 | 1.7 ± 0.3 | 1.9 ± 0.5 | 2.2 ± 0.1 | 2.1 ± 0.1 | 2.1 ± 0.2 | ns | ns | ns |
| Medium Chain | 7 ± 2 | 7 ± 2 | 7 ± 2 | 5 ± 1 | 4 ± 1 | 4 ± 1 | 4.4 ± 0.2 | 4.2 ± 0.2 | 4.3 ± 0.4 | ns | ns | ns |
| C18:1 9c | 4 ± 2 [abc] | 5 ± 2 [ab] | 5 ± 2 [a] | 2 ± 1 [abc] | 1.6 ± 0.3 [bc] | 1.5 ± 0.5 [c] | 1.8 ± 0.1 [abc] | 1.7 ± 0.1 [abc] | 1.8 ± 0.1 [abc] | *** | ns | ns |
| Long Chain | 6 ± 3 | 6 ± 2 | 6 ± 2 | 3 ± 1 | 2.3 ± 0.4 | 2 ± 1 | 4 ± 3 | 2.6 ± 0.1 | 2.7 ± 0.1 | ns | ns | ns |
| TFFA | 27 ± 9 | 27 ± 8 | 27 ± 7 | 17 ± 3 | 14 ± 2 | 15 ± 4 | 16 ± 2 | 13.7 ± 0.1 | 14 ± 1 | ns | ns | ns |

Individual and total free fatty acids of experimental Pecorino Romano DOP cheese manufactured with natural (A and B) and commercial (C) cultures in three different seasons of the cheesemaking year (January, March, and June) in the three factories (1, 2, and 3) involved in the study. For each parameter and cheesemaking season, values within rows (± standard deviation) sharing the different superscript letters differ significantly, according to the ANOVA Tukey–Kramer HSD post hoc test ($p < 0.05$). The statistical evaluation of the overall effects of the culture (C), season (S), and culture x season were evaluated by GLM (general linear model) analysis. *, $P < 0.05$; **, $P < 0.01$; *** $P < 0.001$; ns, not significant.

### 3.5. Sensory Evaluation of Cheese

In each factory (1, 2, and 3) for each cheesemaking season (January, March, and June), CA and CB cheeses were not significantly ($P < 0.05$) different from CC cheeses (Table 8). For the three sensory attributes investigated, odour, taste, and texture, no significant differences were observed, demonstrating that both the natural starters A and B did not change the sensory characteristics of Pecorino Romano cheese.

**Table 8.** Sensory analysis obtained by the difference from control test.

| Factory | Period | Odour | | Taste | | Texture | |
|---|---|---|---|---|---|---|---|
| | | CA vs. CC | CB vs. CC | CA vs. CC | CB vs. CC | CA vs. CC | CB vs. CC |
| 1 | January | 0.3 ± 1.3 n.s. | 0.0 ± 1.2 n.s. | 1.0 ± 0.9 n.s. | 0.3 ± 1.1 n.s. | 0.7 ± 1.5 n.s. | 0.0 ± 0.6 s. |
| | March | 0.6 ± 1.5 n.s. | 0.5 ± 1.7 n.s. | 0.6 ± 1.1 n.s. | 1.1 ± 1.2 n.s. | 0.3 ± 0.9 n.s. | 0.0 ± 0.9 n.s. |
| | June | 0.5 ± 0.9 n.s. | 0.4 ± 0.9 n.s. | 0.7 ± 1.1 n.s. | 0.5 ± 1.1 n.s. | 0.4 ± 0.9 n.s. | 0.2 ± 0.8 n.s. |
| 2 | January | 0.5 ± 1.7 n.s. | 0.7 ± 1.5 n.s. | 1.3 ± 1.5 n.s. | 1.0 ± 1.1 n.s. | 0.5 ± 0.8 n.s. | 1.0 ± 0.9 n.s. |
| | March | 0.4 ± 1.5 n.s. | 0.1 ± 0.5 n.s. | 0.2 ± 0.7 n.s. | 0.6 ± 1.1 n.s. | 0.4 ± 1.5 n.s. | 0.5 ± 0.7 n.s. |
| | June | 0.4 ± 1.1 n.s. | 0.7 ± 1.3 n.s. | 0.3 ± 0.9 n.s. | 0.1 ± 0.8 n.s. | 0.1 ± 0.7 n.s. | 0.1 ± 0.6 n.s. |
| 3 | January | 1.1 ± 1.5 n.s. | 0.3 ± 0.7 n.s. | 0.7 ± 1.2 n.s. | 0.2 ± 0.5 n.s. | 0.8 ± 0.9 n.s. | 0.5 ± 0.8 n.s. |
| | March | 0.3 ± 1.1 n.s. | 0.2 ± 0.6 n.s. | 0.1 ± 0.5 n.s. | 0.3 ± 0.6 n.s. | 0.7 ± 1.1 n.s. | 0.9 ± 1.2 n.s. |
| | June | 0.4 ± 0.7 n.s. | 0.1 ± 0.5 n.s. | 0.1 ± 0.7 n.s. | 0.2 ± 0.6 n.s. | 0.1 ± 0.9 n.s. | 0.2 ± 0.9 n.s. |

For each of the three factories (1, 2, and 3) and each of the three cheesemaking seasons (January, March, and June), the differences ± standard deviation, between CA or CB versus CC was reported for the following sensory aspects: odour, taste, and texture. Data were statistically evaluated by analysis of variance (ANOVA). n.s., not significant.

## 4. Discussion

In the last decades, the improvement of the hygienic milking and cheesemaking conditions [24], together with the heat-treatment of milk required for the reduction of the anti-dairy microorganisms and potentially pathogenic charge [25], has resulted in the depletion of useful natural lactic microflora. This has often made difficult the obtainment of a *scotta-innesto* with an adequate concentration of starter microflora for Pecorino Romano PDO cheese manufacturing, whose product specification requires the use of "native milk ferment cultures from the production area" [8]. For this reason, in recent years, the cheese producers have shown the need to have at their disposal natural starter cultures, in an easy-to-use form (i.e., lyophilised), respecting the characteristics of autochthony requested by the product specification [7,8]. At present, the lyophilised commercial selected starter cultures used are constituted by a few high-concentrated microbial species/strains, generally two to four, selected for their ability to lead the acidification. On one hand, the use of these cultures as indirect inoculum, permitted by the product specification to implement the microbial charge of the *scotta-innesto*, has the principal advantage of guaranteeing the standardisation of a correct acidification performance during Pecorino Romano PDO cheese manufacturing [13]. On the other hand, the low microbial biodiversity that characterises commercial selected cultures, after a long time of intensive use, could affect the diversity of the microbial communities colonizing the production environment and thus milk and cheese biodiversity [26,27]. Moreover, selected cultures may suffer from bacteriophage sensitivity, resulting in slow or incomplete fermentation [28,29], thus requiring cultures rotation [30,31]. This is the reason why Factory 2 used different commercial starters during the three seasons of this experimental cheese-making year. Indeed, *L. helveticus*, not present in A and B collected in the 1960s nor in other natural *scotta-innesto* from the island of Sardinia before the massive use of commercial selected starters [32,33], was present in the SiC used in January, but not in those used in March and June. Natural cultures, unlike the commercial selected ones, are made of an indefinite number of species and strains that are able to mutually replace each other in metabolic pathways, and, besides the advantage of having high bacteriophage tolerance, could help to maintain microbial diversity and improve the sensory richness of products, strengthening their link to the territory of production [1,5]. However, the main drawback attributable to natural cultures is the continuous changing in the microbial communities during daily propagation at the cheese factory [1], which could lead to uncontrolled shifts in their composition that may

affect their technological performances [9]. To overcome this problem, the propagation of natural cultures can be performed, under controlled laboratory conditions, always starting from the original ones (in frozen or lyophilised form) [10,12], favouring the standardisation of microbial performances required by industrial production and preserving the peculiar characteristics of typical products.

The half-century preserved A and B natural starter cultures for Pecorino Romano PDO were recently investigated for their microbial composition, at the strain level, revealing high biodiversity since these cultures are constituted by several strains for each microbial species (*S. thermophilus*, *L. delbrueckii* subsp. *lactis*, *E. faecium*, and *L. reuteri*) [11]. In particular, the recent work carried out by Chessa et al. [11] revealed that A was composed of 5 *Streptococcus thermophilus*, 18 *Lactobacillus delbrueckii* subsp. *lactis*, and 36 *Enterococcus faecium* biotypes, whereas B included 5 *S. thermophilus*, 18 *L. delbrueckii* subsp. *lactis*, 21 *Enterococcus faecium*, and 5 *Limosilactobacillus reuteri* biotypes. In this study, the biodiverse natural cultures were tested for their technological, chemical, and sensory performances in Pecorino Romano PDO cheesemaking at the industrial scale. The cultures were industrially lyophilised and used as indirect inoculum for *scotta* to produce the *scotta-innesto* for cheese manufacturing in three Sardinian cheese factories, and the cheeses obtained were compared, by a multidisciplinary approach, to those manufactured using commercial selected starters (different commercial cultures for each of the three factories). The microbial variability found between *scotta/siero-innesto* obtained inoculating natural and commercial starters regarded mainly the presence of enterococci, not found in SiC (prepared using the commercial selected starters) but always detected in SiA and SiB. This microbial group belongs to non-starter lactic acid bacteria (NSLAB), involved in cheeses' aroma development during ripening, conferring typical flavour to cheeses due to their proteolytic and lipolytic activity [34]. Another indicator of the differences between natural and commercial selected starters was the pH of the *scotta/siero-innesto*, lower in the SiCs. Furthermore, different concentrations of total microbial population and the main microbial groups (thermophilic cocci and bacilli) of the three *scotta/siero-innesto* (SiA, SiB, and SiC) were observed, as well as the concentration of enterococci, which also varied among the cheesemaking seasons. Although the application of natural or commercial selected starters for the preparation of the *scotta/siero-innesto* generally did not influence the Pecorino Romano cheese microbiota and the gross composition of the six-month-ripened cheeses, which were instead influenced mostly by the seasonality, the use of lyophilised natural cultures could benefit the cheesemaking process. In fact, commercial selected starters are allowed to be used only as indirect inoculum to integrate the natural microflora already present in the *scotta*, while natural cultures, such as A and B *scotta-innesto*, being biodiverse and autochthonous, even if in a different physical status (i.e., in lyophilised form), could be directly inoculated in the milk vat. At the moment, this practice is not specified by the Pecorino Romano PDO product specification [8], and the need for an update of the operational Control Plan [13] should be evaluated. However, since the concentration of the lyophilised cultures used in this study was not adequate for an effective inoculum in milk, their overnight incubation in *scotta* was unavoidable. To overcome this problem, a parallel trial [12] was focused on methods for the improvement of the concentration of bacterial cells in the lyophilised A and B cultures, preserving the microbial communities' balances. The increase in microbial cells concentration could allow the direct inoculum with a labour-saving advantage for the Pecorino Romano cheesemakers, simplifying long and annoying procedures.

From a chemical point of view, in general, no effect due to the type of *scotta/siero-innesto* used was observed for the physico-chemical and main gross-composition parameters of experimental Pecorino Romano cheeses, except for the moisture content. Cheeses from Factory 1 and 2 showed a lower moisture content when produced with the commercial selected starter (CC) instead of the cultures belonging to the Agris BNSS collection. This behaviour was particularly evident in the spring and summer products. In general, the cheesemaking season significantly affected the gross-composition parameters of the experimental cheeses. In particular, cheese produced in the spring period (March) had a lower fat

content than winter (January) and summer (June) cheeses. This behaviour was due to the natural variation of the protein/fat ratio in sheep milk, which increases from November to April and then decreases until July [35]. The values of gross-composition parameters were comparable to those observed in Pecorino Romano PDO cheese at 5–8 months of ripening, with the exception of the NaCl content, which, with respect to the variability range, was lower in the cheese from Factory 1 and higher in cheeses from Factory 2 and 3, respectively [36]. The proteolytic indices, which generally were not affected by the starter culture used or the cheesemaking season, were within the range of variability observed in Pecorino Romano PDO cheese (7–8 months of ripening) [36]. Cheeses produced in Factories 2 and 3 featured a lower (although not significant) NS/NT content than those from Factory 1, probably due to the higher salt content (ranging from 4.8% to 6%) that characterised these cheeses and that slowed down the enzymatic processes of cheese ripening. The *scotta/siero-innesto* used did not affect the lipolytic ripening process of the experimental cheeses (CA, CB, and CC), which was weakly influenced by the cheesemaking season and more pronounced in cheeses produced in the winter period (Table 7). The contents of individual and total fatty acids in the cheeses produced in the three factories differed from each other, probably due to the use of different rennet paste, and they were comparable to the lower limit of the variability range observed in Pecorino Romano PDO cheese (7–8 months of ripening) [36]. This behaviour was probably due to the qualitative decay and the high variability of pregastric lipase content that characterises the rennet paste currently present on the market and used in cheesemaking, regulating the lipolytic ripening process in Pecorino Romano cheese [36].

The sensory characteristics of cheese derive from a combination of complex physicochemical and microbiological factors closely linked to the raw material and production technology [37]. Previous studies showed that, in Pecorino Romano PDO, these factors are strongly influenced by the NaCl content [36,38], which has a direct impact on the sensory characteristics of this product [39]. In particular, the highest NaCl content that characterises Pecorino Romano cheese (from 2% up to 6%), compared to other hard-ripened cheeses (Parmigiano Reggiano 1.6%, Grana Padano 1.5%, and Pecorino Sardo 1.9%), could play a suppression role over the other basic tastes and an inhibiting effect on microorganisms, responsible for the main cheeses' flavor compound production. The same authors found, in a previous study, that NaCl content above 3% significantly affects the taste, aroma, and structure of Pecorino Romano PDO, negatively influencing the consumers' liking [39,40]. Further studies should be conducted on this topic, but it seems that this NaCl content value could represent a sensory terminal threshold in Pecorino Romano cheese. Therefore, also in this work, the high salt content may have affected the sensory characteristics of Pecorino Romano PDO, masking the effects of the different starters used. Although the NaCl content was significantly different in cheeses produced in Factory 1 in June, this was not perceived by the panelists, since all the salt concentrations were always much higher than 3%.

## 5. Conclusions

The study concerning the use of the biodiverse natural starter cultures in Pecorino Romano PDO cheesemaking highlighted that, in the ripened cheeses, only differences attributable to the season of production were observed with the culture-dependent techniques applied. Further studies focusing on the strain biodiversity of the cheese microflora would be needed by associating culture-independent methods in order to widen the view on the microbial biodiversity of both starter cultures and cheeses. The results obtained proved that natural starter cultures, reproduced and preserved in an easy-to-use freeze-dried form, can be used in industrial-scale production, ensuring high stability in the technological performances and preserving the microbial, chemical, and sensory characteristics of Pecorino Romano PDO cheese. Furthermore, the large advantage of using a lyophilised natural culture (e.g., A and B) instead of a commercial selected one is the possibility to add such a kind of starter directly to the milk in the vat, bypassing the time-consuming and

risky practice of daily managing the culture for the preparation of the *scotta-innesto* in the fermenter, also with a view to saving energy.

**Supplementary Materials:** The following are available online at https://www.mdpi.com/article/10.3390/su13158214/s1: Table S1: Technological cheesemaking parameters used in the Factories.

**Author Contributions:** Conceptualization, L.C., A.P., E.D. and R.C.; methodology, L.C., A.P., E.D., I.D., C.P., R.D.S., M.A. and R.C.; formal analysis, L.C., A.P., E.D., I.D., C.P. and M.A.; investigation, L.C., A.P., E.D., I.D., C.P., R.D.S. and M.M.; data curation, L.C., A.P., E.D., I.D., C.P., R.D.S., M.A. and R.C.; writing—original draft preparation, L.C., A.P., C.P., R.D.S., M.A. and R.C.; writing—review and editing, L.C., A.P., E.D., I.D., C.P., R.D.S., M.A. and R.C.; supervision, L.C. and R.C.; project administration, R.C. All authors have read and agreed to the published version of the manuscript.

**Funding:** This research was supported by the Regione Autonoma della Sardegna — POR-FESR 2014–2020 – CLUSTER Project called 'La diversificazione di prodotto nell'ambito del Pecorino Romano DOP' (Det. n. 1–May 3, 2016, Sardegna Ricerche).

**Institutional Review Board Statement:** Not applicable.

**Informed Consent Statement:** Not applicable.

**Data Availability Statement:** Not applicable.

**Acknowledgments:** The authors thank Maria Carmen Fozzi for her precious contribution to the microbiological analyses.

**Conflicts of Interest:** The authors declare no conflict of interest.

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
