# Peer review of "Autochthonous Natural Starter Cultures: A Chance to Preserve Biodiversity and Quality of Pecorino Romano PDO Cheese"

_sustainability, doi:10.3390/su13158214_

Round 1

Reviewer 1 Report

The manuscript is well written, easy to follow and is based on experiments done in real production conditions. The novelty is in re-introduction of more than half of century-old bacterial composition to restore the original pecorino starters and to adapt it to the state-of-the-art technology and contemporary food production hygiene. Despite that, there were some questions rising from manuscript.

Explanations needed:

Line 274-275 claims, that no effect on scotta-innesto microbial content was observed in Factory 3. According to authors (lines 104, 248, and supplementary table S1) in Factory 3 only siero was used! Was there an error in explanation?

In lines 400-401 authors claim, that „In Factory 1, cheeses produced in June with natural starter B showed a NaCl content significantly higher than other cheeses.“  The similar trend (statistically seems to be not significant, but still the same average values in table S2) is seen for Factory 3 in June also. Authors could give some speculations about it. The salt content is also affecting the sensory characteristics and 6.0 percent of salt in both Factories in June with CB are twice of recommended salt content value for ensure the customer preference (look at lines 544-555).

Corrections needed:

Lines 154-155 should appear as: „Microbial counts were expressed as average values ± standard deviation (SD) Log CFU/mL or Log CFU/g. “

Author Response

Answers to the Reviewer 1 comments

The manuscript is well written, easy to follow and is based on experiments done in real production conditions. The novelty is in re-introduction of more than half of century-old bacterial composition to restore the original pecorino starters and to adapt it to the state-of-the-art technology and contemporary food production hygiene. Despite that, there were some questions rising from manuscript.

Explanations needed:

Question: Line 274-275 claims, that no effect on scotta-innesto microbial content was observed in Factory 3. According to authors (lines 104, 248, and supplementary table S1) in Factory 3 only siero was used! Was there an error in explanation?

Answer: Scotta-innesto was erroneously indicated since in Factory 3 only siero was used. Therefore, scotta-innesto was replaced with siero-innesto in line 275 of the revised manuscript version.

Question: In lines 400-401 authors claim, that „In Factory 1, cheeses produced in June with natural starter B showed a NaCl content significantly higher than other cheeses.“  The similar trend (statistically seems to be not significant, but still the same average values in table S2) is seen for Factory 3 in June also. Authors could give some speculations about it. The salt content is also affecting the sensory characteristics and 6.0 percent of salt in both Factories in June with CB are twice of recommended salt content value for ensure the customer preference (look at lines 544-555).

Answer: In Factory 1 and 3 the trend of NaCl content in CB cheese was similar (6%) but while in Factory 1 the differences were significant, in Factory 3 the different NaCl content among was not. This was due to the differences in NaCl concentration (6.0 vs 4.8 and 5.2) found in Factory 3, smaller than those found in Factory 1 (6.0 vs 4.2 and 4.1) (Table S2). The paragraph was modified in order to better explain the results obtained (lines 402-404).

NaCl content above 3% does not ensure the consumer preferences but, conversely, affect their liking for the product. Indeed, as reported by Di Salvo et al., (reference 39) that investigated the effects of salt content reduction in Pecorino Romano PDO cheese on sensory attributes and consumer perception, the acceptability values of consumers increased as the NaCl content decreased (from 4% to 2%), and salt amount significantly influenced the intensity of the sensory attributes specifically linked to flavour and texture of the product.

The paragraph was modified (lines 550-555).

Corrections needed:

Question: Lines 154-155 should appear as: „Microbial counts were expressed as average values ± standard deviation (SD) Log CFU/mL or Log CFU/g. “

Answer: the sentence was modified as suggested.

Reviewer 2 Report

Your research is an important contribution in the enhancement of territorial productions and, even more so, if certified (PDO).

Author Response

Answers to the Reviewer 2 comments

Your research is an important contribution in the enhancement of territorial productions and, even more so, if certified (PDO).

The authors would like to thank the Reviewer for the positive comments about the study

Reviewer 3 Report

Comments to the Author

MANUSCRIPT DETAILS

Ms. Ref. No.: sustainability-1292157

Title: Autochthonous natural starter cultures: a chance to preserve biodiversity and quality of Pecorino Romano PDO cheese

Article Type: Research Article

JOURNAL: Sustainability

GENERAL COMMENTS

This study investigated the effect of using three autochthonous starter cultures (SR30, SR56, and SR63) combined into two starter mixes, in production of Pecorino Romano cheese comparing with commercial starters.

The interest in this manuscript is significant enough to merit publication

My recommendation on submitted manuscript to Sustainability is to be accepted after major revisions.

The comments and questions provided below may help the authors to put the manuscript into better appropriate form for publication.

SPECIFIC COMMENTS

- Language in manuscript need to be revised thoroughly

- What did authors mean by “natural” culture? Did you mean isolated wild cultures from cheese or dairy sources? As “commercial” starters are also “natural”.

- Kindly clarify the average values, replicates, significance letters, SD and abbreviations in all figures’ captions and tables’ footnotes.

- Why the control starter was not applied for the 3rd factory? Even if “the Factory 3 had only two fermenters”, as authors mentioned, which could be managed for comparison reasons.

- Chemical characteristics of cheese (including fatty acids profile) Tables S2 and S4 are important data that should be included within main manuscript.

- Missed sensory evaluation data should be exhibited, even with no significance.

- I suggest re-arranging data illustrations to exhibit the main data within the main manuscript in parallel with journal Authors Guidelines.

Author Response

GENERAL COMMENTS
This study investigated the effect of using three autochthonous starter
cultures (SR30, SR56, and SR63) combined into two starter mixes, in
production of Pecorino Romano cheese comparing with commercial starters.
The interest in this manuscript is significant enough to merit publication
My recommendation on submitted manuscript to Sustainability is to be accepted
after major revisions.
The comments and questions provided below may help the authors to put the
manuscript into better appropriate form for publication.

SPECIFIC COMMENTS

Question: Language in manuscript need to be revised thoroughly

Answer: the manuscript was revised in the English, and all changes were performed using the MS Office Track Changes tool.

Question:  What did authors mean by “natural” culture? Did you mean isolated wild
cultures from cheese or dairy sources? As “commercial” starters are also
“natural”.

Answer: the definition of “natural” cultures is referred to complex and undefined microbial consortia consisting of an indefinite number of species and strains, starter and non-starter, coexisting in equilibrium not reproducible in a place other than of their origin. This concept is opposed to that of “selected” cultures, composed by few species/strains isolated and selected under laboratory conditions then put together basing on the desired properties.

This concept is well known and reported in several studies (Gatti et al., 2014, doi:10.3168/jds.2013-7187; Bassi et al., 2015, doi:10.1016/j.cofs.2015.03.002; Campus et al., 2018, doi:10.3389/fmicb.2018.00617).

Usually, commercial starter cultures, such as those allowed for the production of Pecorino Romano PDO cheese, sold by biotech companies, are selected cultures. Therefore, in this study, the technological performances of natural starter cultures belonging to the BNSS Agris microbial collection were compared to commercial selected starter cultures.

To better clarify the concept, the adjective “selected” was added in the text associated to “commercial”.

Question:  Kindly clarify the average values, replicates, significance letters, SD and
abbreviations in all figures’ captions and tables’ footnotes.

Answer: All the figures captions and tables footnotes were improved as suggested.

Question:  Why the control starter was not applied for the 3rd factory? Even if “the
Factory 3 had only two fermenters”, as authors mentioned, which could be
managed for comparison reasons.

Answer: actually, the control starter was applied also in Factory 3. Indeed control cheeses were obtained and compared to those obtained using A and B cultures. As noted by the Reviewer, only two fermenters were available for this experimental trial, and they were used for the two cultures A and B. To overcome this problem and obtain the control cheeses on the same working day, so from the same milk, the commercial starter, normally used as indirect inoculum, was inoculated directly to the milk, at a concentration comparable to that of the natural ones.

Question:  Chemical characteristics of cheese (including fatty acids profile) Tables
S2 and S4 are important data that should be included within main manuscript.

Missed sensory evaluation data should be exhibited, even with no
significance.

Answer: Table S2, S3, and S4 were moved into the main text and now are presented as Table 5, Table 6, and Table 7.

Moreover, Table 8 reporting the sensory data was added to the text.

Question:  I suggest re-arranging data illustrations to exhibit the main data within
the main manuscript in parallel with journal Authors Guidelines.

Answer: The main data were included in the text, as suggested, and the figures numbers updated accordingly.

Round 2

Reviewer 3 Report

Comments to the Author

MANUSCRIPT DETAILS

Ms. Ref. No.: sustainability-1292157

Title: Autochthonous natural starter cultures: a chance to preserve biodiversity and quality of Pecorino Romano PDO cheese

Article Type: Research Article

JOURNAL: Sustainability

GENERAL COMMENTS

In the revised manuscript sustainability-1292157, the authors have addressed the comments raised by reviewer and presently meet the requirements and standards of Sustainability. The data were illustrated in a better organized manner.